



# The HYPERMAQ dataset

Héloïse Lavigne[1], Ana Dogliotti[2], David Doxaran[3], Fang Shen[4], Alexandre Castagna[5], Matthew Beck[1], Quinten Vanhellemont[1], Xuerong Sun[4], Juan Ignacio Gossn[2,7], Renosh Pannimpullath[3], Koen Sabbe[5], Dieter Vansteenwegen[6], Kevin Ruddick[1]

[1] Royal Belgian Institute of Natural Sciences, Brussels, Belgium
[2] Instistuto de Astronomía y Física del Espacio (IAFE), CONICET-Universidad de Buenos Aires, Buenos Aires, Argentina
[3] Laboratoire d'Océanographie de Villefranche, UMR7093 Sorbonne Université /CNRS, Villefranche-sur-Mer, France
[4] State Key Laboratory of Estuarine and Coastal Research (SKLEC), East China Normal University, Shanghai, China
[5] Laboratory of Protistology and Aquatic Ecology, Ghent University, Ghent, Belgium
[6] Flanders Marine Institute (VLIZ), Ostend, Belgium
[7] European Organisation for the Exploitation of Meteorological Satellites (EUMETSAT), Darmstadt, Germany.

*Correspondence to*: Héloïse Lavigne (hlavigne@naturalsciences.be)

**Abstract.** Because of the large diversity of case 2 waters ranging from extremely absorbing to extremely scattering waters and the complexity of light transfer due to external terrestrial inputs, retrieving main biogeochemical parameters such as chlorophyll-a or suspended particulate matter concentration in these waters is still challenging. By providing optical and biogeochemical parameters for 180 sampling stations with turbidity and chlorophyll-a concentration ranging from 1 to 700 FNU and from 0.9 to 180 mg m$^{-3}$ respectively, the HYPERMAQ dataset will contribute to a better description of marine optics in optically complex water bodies and can help the scientific community to develop algorithms. The HYPERMAQ dataset provides biogeochemical parameters (i.e. turbidity, pigment and chlorophyll-a concentration, suspended particulate matter), apparent optical properties (i.e. water reflectance from above water measurements) and inherent optical properties (i.e. absorption and attenuation coefficients) from six different study areas. These study areas include large estuaries (i.e. the Rio de la Plata in Argentina, the Yangtze Estuary in China and the Gironde Estuary in France), inland (i.e. the Spuikom in Belgium and Chascomùs lake in Argentina) and coastal waters (Belgium). The dataset is available from Lavigne et al (2022), https://doi.pangaea.de/10.1594/PANGAEA.944313.

## 1 Introduction

In marine optics, certain water properties such as the concentration of chlorophyll-a ([Chl-a] hereafter) or suspended particulate matter (SPM hereafter) are inferred from water leaving reflectance allowing a powerful satellite based monitoring. However, although algorithms are well matured in clear case 1 waters (Morel and Prieur, 1975; Morel and Maritorena, 2001), it is not the case in optically complex case 2 waters where apparent optical properties (AOPs) and inherent optical properties (IOPs) are influenced not only by [Chl-a] but also by terrestrial optically-active substances such as suspended sediments and colored dissolved organic matter (CDOM) that do not covary



with [Chl-a]. Given the complexity of light transfer in these waters and the large diversity of case 2 waters, algo-
rithm definition is much more challenging (Odermatt et al., 2012) and requires datasets covering the extreme
variability of case 2 water conditions. Hence, any additional data obtained in optically complex waters are valuable
to the scientific community as they will help to better understand marine optics in such waters and to design ocean
color algorithms.
The present dataset (Lavigne et al., 2022; https://doi.pangaea.de/10.1594/PANGAEA.944313) has been collected
as part of the HYPERMAQ project.  During this project, different types of optically complex waters with turbidity
ranging from moderate to extreme (1 to 700 FNU) and [Chl-a]  ranging from low to very high (0.9 to 180 mg m$^{-3}$)
have been sampled in various locations around the world. Their main optical and biogeochemical parameters
are shared in this dataset, including measurements of water-leaving reflectance, turbidity, non-water light absorp-
tion and attenuation coefficients as well as SPM, [Chl-a] and other pigment composition. In the next sections,
study areas, sampling methodology and final HYPERMAQ datasets are described.
**2 Sites**
Different sites, generally characterized as optically complex case-2 waters with a turbidity and [Chl-a] ranging
from moderate to extremely high, have been sampled in coastal and inland waters in Belgium, France, Argentina
and China (Figure 1).
**2.1 Belgian coastal waters**
The Belgian coastal waters (latitudes:51.27° to 51.59°N; longitudes: 2.50° to 3.15°E) have been sampled in April
2018 and July 2018 from the RV Simon Stevin (Table 1). Belgian coastal waters are dominated by Atlantic waters
which enter from the English Channel (Lacroix et al., 2004) and experience very strong along shore tidal currents
which cause sediment resuspension leading to high turbidity. SPM concentrations range from less than 1 g m$^{-3}$ in
offshore and deeper waters to more than 100 g m$^{-3}$ in very shallow waters. Phytoplankton blooms, characterized
by high [Chl-a] concentration (more than 10 mg m$^{-3}$), develop in spring from March to May. During summer, the
biomass remains rather high (5 to 10 mg m$^{-3}$) compared to winter when phytoplankton growth is mostly limited
by light (Lancelot et al., 2005). The blooming season is mostly dominated by two taxa: diatoms in early spring and
summer and *Phaeocystis globosa* in April-May (Muylaert et al., 2006).
**2.2 Spuikom lagoon**
The Spuikom lagoon (latitude: 51.23°N, longitude: 2.95°E) is an artificial basin that is connected to Ostend harbor
(Belgium) by a lock system. The Spuikom has a surface area of 0.82 km$^2$ and an average depth of 1.5 m. In the
past, it has been used as a flushing basin to flush sediments from the harbor channel. Today it is used for leisure
and commercial activities like sailing and shellfish farming. The Spuikom can experience events of phytoplankton
blooms, of high turbidity (when strong winds cause the resuspension of bottom sediments), and of clear waters,
which allow the development of microphytobenthic biofilms and macroalgae in the bottom (Castagna et al., 2022).



The system was sampled during the growth season of 2018 (April and July, Table 1). Measurements were
performed from inflatable boats provided by Ghent University and VLIZ (Zeekat).

**2.3 Gironde Estuary**

The Gironde Estuary, southwest France, is a good example of sediment-dominated case 2 waters influenced by
river inputs. The Gironde Estuary has been sampled between 17-20 September 2018 in two locations: Pauillac
(latitude: 45.1975°N, longitude: -0.7422°E) close to the maximum turbidity zone and Le Verdon (latitude:
45.5438°N, longitude: -1.042°E) close to the river mouth.  In the Gironde Estuary, the origin of the particles is
twofold: inputs from rivers Garonne and Dordogne and erosion of recently settled sediments by tidal currents
(Castaing and Allen, 1981). The suspended matter is a mixture of organic and mineral composites, where the
organic fraction represents less than 2% of the total material (Doxaran et al. 2002). The mineral fraction is com-
posed of micas (63%) and quartz (25%), while clay phases contain four minerals: montmorillonite (30%), illite
and interstratified minerals (40%), kaolinite (15%), chlorite and interstratified minerals (15%). [Chl-a] and CDOM
concentrations are low, with [Chl-a] ranging from 1 to 3 mg m$^{-3}$ (Irigoien & Castel, 1997), and dissolved organic
carbon (DOC) ranging from 1 to7 mgC L$^{-1}$ (Abril et al., 1999; Castaing and Allen, 1981). The Gironde Estuary
has well-developed turbidity maximum zones, with both tidal asymmetry and density residual circulation involved
in their formation (Castaing and Allen,1981). It is characterized by SPM concentrations ranging from 10 to 1000
g m$^{-3}$ within surface waters (Doxaran et al. 2009a).

**2.4 Chascomús lake**

Chascomús lake, located in the Pampa Plain in the Buenos Aires Province in Argentina (latitude: -35.5828°N,
longitude: -58.0202°E), with a surface area of ~ 30 km$^2$, is a highly turbid, shallow lagoon (average depth of ~ 1.9
m), permanently mixed due to intense and persistent winds (Torremorell et al., 2007). Total suspended matter
varies widely from 66.3 to 614 g m$^{-3}$ with a mean value of 227.3 ±133.7 g m$^{-3}$ (Diovisalvi et al. 2014) and on
average the inorganic content represented ~65%. Nephelometric turbidity also widely ranged from 76.46 to 509.74
NTU, with a mean value of 209.18±112.76 NTU. Turbidity was highly correlated to SPM while no significant
correlation with [Chl-a] was found (Pérez et al. 2011). Total [Chl-a] concentration ranged from 50.6 to 856.3 mg
m$^{-3}$ (mean = 328.5 ±173.4 mg m$^{-3}$) during the 2005-2009 sampled period (Diovisalvi et al. 2014). The lake is
characterized by high primary production (Torremorell et al., 2009) and a rich and diverse phytoplankton commu-
nity, mostly composed of cyanobacteria. In terms of biovolume, cyanobacteria contribute 50% to total phytoplank-
ton biovolume and 75% to total C in the water column (Diovisalvi et al. 2010). Despite the high CDOM absorption
($a_{CDOM}$, mean $a_{CDOM}$(440)=4.65 ± 0.91 m$^{-1}$), absorption by particulate fraction ($a_p$) has a prominent role in light
absorption, for which both phytoplankton pigments ($a_{phy}$) and non-pigmented particles ($a_{NAP}$) contribute similarly
to total particulate absorption (Pérez et al. 2011). Both SPM (especially the inorganic part) and [Chl-a] (less pro-
nounced) show seasonal variation with increasing values in spring and summer (Mid-September to Mid-March),
while the dissolved fraction did not show a significant seasonal difference (Pérez et al. 2011). The HYPERMAQ
field campaign in Chascomús lake took place on 9-10 April 2018. Radiometric, in-water measurements as well as
samples were collected at the end of a 164 m long pier.



### 2.5 Río de la Plata

The Río de la Plata is a large and shallow funnel shaped estuary with high values SPM, ranging from 100 to 300 g m$^{-3}$ (Framiñan and Brown, 1996) and reaching 940 g m$^{-3}$ in the maximum turbidity zone (Dogliotti et al. 2014). Turbidity values widely vary between 2 and 680 FNU (Dogliotti et al. 2016). SPM, turbidity and [Chl-a] spatial distribution and temporal variability is highly variable. In the upper estuary, a freshwater with tidal regime area, turbidity increases from January to April/May (with higher values along the southern Argentinian coast of compared to the northern Uruguayan coast), and decreases from June to September (Dogliotti et al. 2016). In turn [Chl-a] also show high spatial variability, in the upper estuary higher values are generally found in the northern part (Uruguay) compared to the southern part (Argentina). In particular, high [Chl-a] have been recorded during spring-summer months related to cyanobacteria blooms both along the Uruguay (Aubriot et al. 2020) and Argentine (Dogliotti et al. 2021) coasts, when [Chl-a] values as high as 13.6 and 153 mg m$^{-3}$ have been recorded, respectively. Measurements in the Rio de la Plata were performed from a fixed 500 m long pontoon at the Palermo Pescadores Club in Buenos Aires (latitude: -34.5609°N, longitude: -58.3988°E) on 4 and 5 April 2018.

### 2.6 Yangtze Estuary

The Yangtze Estuary is located on the east coast of China and close to East China Sea (Figure 1). Influenced by the Yangtze River, the largest river in China and the third largest in the world, which discharges an annual average of $9\times10^{11}$ m$^3$ of freshwater and $4\times10^8$ tons of sediment into the estuary (Chen et al., 2003), the Yangtze Estuary is an extremely turbid area (Shen et al., 2010a). Taking 2009 as example, the annually averaged of SPM in surface waters varied from 58 g m$^{-3}$ at the upstream limit of the estuary to about 600 g m$^{-3}$ at the mouth area, and fell again to 57 g m$^{-3}$ at the seaward limit of fresh water diffusion (Li et al., 2012). Due to the different river discharges, the SPM of the Yangtze Estuary exhibits seasonal variations (Shen et al., 2013), with SPM in the upper estuary (lower estuary) during flood season significantly higher (lower) than that during the dry season. Over the past 37 years, SPM in Yangtze Estuary demonstrated an overall declining pattern (Luo et al., 2022), with SPM in the inner estuary responding most promptly (40.3% reduction) after the operation of Three Gorges Dam. [Chl-a] also shows seasonal variations in Yangtze Estuary, ranging from 0.03 to 3.10 mg m$^{-3}$ and from 0.88 to 31.5 mg m$^{-3}$ during spring and summer seasons of 2008, respectively (Shen et al., 2010b). In addition, the Yangtze Estuary is an area with frequent outbreaks of algal blooms, with diatoms being the most frequently reported group (Shen et al., 2019; Zhu et al., 2019).

Two Hydrological Stations in Chongming Island, Shanghai, China, namely Chongxi (longitude:121.193°E, latitude:31.759°N) and Baozhen (longitude:121.609°E, latitude:31.520°N) have been sampled from 30 May to 8 June in 2018 (Table 1).

### 3. Data collection

The dataset contains measurements of the turbidity and, if available, concomitant SPM, absorption and attenuation coefficients, [Chl-a] and reflectance measurements are also included (Lavigne et al., 2002; https://doi.pangaea.de/10.1594/PANGAEA.944313). An overview of the dataset, with the number of observations after quality

control for each site and parameter, is provided in Table 2. The measurement methodology for each parameter is
described below.
**3.1 Water-leaving reflectance**
Above-water reflectance was determined using three TriOS/RAMSES hyperspectral spectroradiometers, two spec-
troradiometers measure radiance and one irradiance. The same TriOS instruments from RBINS institute were used
for all campaigns except the ones which occurred in Argentina where only instruments from IAFE institute were
available. The spectrometers measure in the 350-950 nm range with a sampling interval of 3.3 nm and effective
spectral resolution of 10 nm. The instruments were mounted on a frame and placed in the bow of the vessels
(Belgian coastal zone and Spuikom) or fixed to a rail when measurements were made from pontoons (Gironde,
Chascomús and Rio de la Plata). Zenith angles of the sea- and sky-viewing radiance sensors were set to 40°. Prior
to each measurement, the azimuth angle of the sensors was adjusted to obtain a relative azimuth angle with respect
to the sun of 90°, either left or right to get the best unobstructed view of the water and minimize structure pertur-
bation when measuring from pontoons. Simultaneous upwelling water radiance ($L_u$), downwelling sky radiance
($L_{sky}$) and downwelling irradiance ($E_d$) were collected every 10 s for 10 min. Data was acquired using MSDA-XE
software and radiometrically calibrated using the latest calibration update from annual laboratory calibration. Wa-
ter reflectance ($\rho_w$) was calculated following
$$\rho_w(\lambda) = \frac{L_u(\lambda) - \rho_{sky}L_{sky}(\lambda)}{E_d(\lambda)}\pi$$

Where $\rho_{sky}$ is the air-sea interface reflection coefficient which is calculated based, when available, on wind speed
as in Ruddick et al. (2006) or set to a fixed value of 0.0256 when measured in estuaries from fixed pontoon con-
sidering that surface waves are fetch-limited and not related to wind speed. The data processing, including quality
control, are described in Ruddick et al. (2006).
**3.2 Turbidity**
Turbidity was measured with two handheld HACH 2100P/Q ISO turbidimeters from RBINS and IAFE institutions.
In the HYPERMAQ dataset, turbidity data measured with the instrument from IAFE were provided by default as
they cover the most of the campaigns. However, when turbidity data from IAFE instrument were not available
(Belgian coastal waters, April 2018 and Spuikom April 2018), the values obtained with the instrument of the
RBINS were used. Figure 2 shows the good consistency of both instruments ($r^2$=0.99). Water samples were col-
lected from the surface with a bucket or from subsurface with a NISKIN bottle for measurements in coastal waters.
A 10 mL vial was filled and turbidity was determined in Formazin Nephelometric Unit (FNU) with the ratio of
light scattered at 90° compared to the transmitted light at 860 nm. Turbidity was recorded in triplicates and the
median value was used. Turbidimeters were controlled with standards STABCAL Stabilized Formazin Turbidity
of 0.1, 20, 100 and 800 FNU before and after each campaign.
In water turbidity was also measured with an OBS501 (OBS hereafter) using a CR200 data logger. Turbidity
measurements are derived from back-scattering with a field-of-view ranging from 125 to 170 degree and 90-degree
side-scattering of a signal emitted at 850 nm and data are provided in Formazin Backscatter Unit (FBU) and in



Formazin Nephelometric Unit (FNU), respectively. When deployed from a pier, OBS was continuously recording
data at subsurface throughout the whole day and values corresponding to specific stations were extracted from the
time-series in a time-window of 10 minutes centered on the timing of the radiometric measurement and water
sampling. When deployed from a boat, the OBS was maintained at subsurface (1 m depth) for at least 5 minutes.
Then, from a visual check, leading and trailing data of each time-series were removed and the central values were
averaged to obtain a final value.
**3.4 *In situ* absorption, beam attenuation and scattering coefficients**
The underwater absorption- and attenuation-meter (AC-9, WETLabs, Inc.) used was modified to cover the visible
and near-infrared (NIR; 700 to 900 nm) spectral regions. It was designed with three visible (centered at 440 nm,
555 nm and 630 nm) and six NIR (centered at 715 nm, 730 nm, 750 nm, 767nm, 820 nm and 870 nm) spectral
channels, and a short pathlength (10 cm) appropriate for turbid coastal waters. At the sampling sites, the AC-9
sensor was either deployed within the water column using an electrical water pump (SBE, SeaBird, Inc.) or used
as a bench photometer passing the water samples right after collection through the tubes by gravimetry. The AC-
9 data recorded just below the water surface were averaged over the last minute of acquisition to obtain the mean
attenuation and absorption spectra for each station. Temperature and salinity corrections were applied as recom-
mended by the manufacturer. As in Doxaran et al. (2007), the residual scattering effects on absorption measure-
ments were corrected by applying the "proportional" method using 870 nm as the reference wavelength. The scat-
tering coefficient was calculated as the difference between the measured beam attenuation coefficient, $c_{nw}$, cor-
rected for temperature and salinity effects, and the absorption coefficient, $a_{nw}$, corrected for temperature, salinity
and scattering effects. Those attenuation and absorption coefficients were referenced to pure water (non-water,
subscript "nw"), so that the scattering coefficient obtained by difference corresponds to the scattering coefficient
of marine particulates, $b_p$ in m$^{-1}$. Small bubbles can contribute to the measured attenuation and scattering, but in
turbid systems particles dominate the signal. One of the main issues encountered when sampling highly turbid
waters was the saturation of the measured absorption and/or attenuation coefficients, which sometimes occurred
at short visible wavelengths and even in near-infrared bands in the case of extremely turbid waters. This saturation
was easily detected and the corresponding spectra were systematically removed from the dataset.
When possible, after AC-9 data measurements, the water sample collected was directly filtered through disc filters
(pore size 0.2 μm, Whatman). As in Doxaran et al. (2009b), the tube was rinsed twice with Milli-Q water and once
with the filtrate, and then filled with the filtrate. The absorption signal of the filtrate was measured, providing
$a_{CDOM}$ in m$^{-1}$ after applying corrections for temperature and salinity. The absorption coefficient of suspended par-
ticles ($a_p$, in m$^{-1}$) was finally calculated by subtracting the signal from the non-water absorption coefficient.
**3.5 Concentration of Suspended Particulate Matter and Suspended Inorganic Particulate Matter**
SPM concentration was determined gravimetrically following the protocol of Tilstone et al. (2002) which is based
on Van der Linde (1998). Water was sampled from the surface (maximum 2 m depth) with a NISKIN bottle on
board the RV Simon Stevin or with a bucket in estuarine and inland waters. A sufficient volume of water was
filtered on a pre-ashed GF/F filter and conserved at -20°C before analysis. The volume filtrated was determined



as a function of the turbidity following recommendations of Neukermans et al. (2012). Inorganic suspended par-
ticulate matter (SPIM) was also calculated in all campaigns except in the Yangtze river. All the SPM measurements
have been conducted with 3 replicates to assess variability except for the campaigns in the Yangtze Estuary where
only one sample has been measured per each station. Filters were dried at 75°C for 24 hours and weighed in order
to determine the suspended matter concentration (SPM, in g m$^{-3}$). For SPIM measurements, filters were then
burned at 450°C for 4 hours to remove the organic part, and weighed again to estimate the suspended inorganic
particulate concentration (SPIM, in g m$^{-3}$).

**3.6. Chlorophyll-*a* and other pigment concentrations**

[Ch*a*] and phytoplankton pigments were determined using High Performance Liquid Chromatography (HPLC)
following the protocol of Van Heukelem and Thomas (2001) in campaigns in the Belgian coastal waters, the Spu-
ikom and the Gironde. In Belgian waters, measurements were provided by the LifeWatchBE sampling campaigns
(Mortelmans et al., 2019, Flanders Marine Institute, 2021) of VLIZ. Pigment standards were acquired from the
Danish Hydrographic Institute (DHI). In the Gironde, the analysis of pigments were performed by the SAPIGH
analytical plateform of the "Institut de la Mer de Villefranche" (CNRS-France). In the Argentinian campaigns
[Chl-a] was determined spectrophotometrically using hot ethanol (60-70 °C) (Jespersen and Christoffersen 1987).
As for turbidity and SPM, water samples have been collected from surface water with a bucket in inland waters or
subsurface waters with a NISKIN in sea water.

**4. Results and discussions**

**4.1 SPM and turbidity results**

In the HYPERMAQ dataset SPM ranges between 1 g m$^{-3}$ and 474 g m$^{-3}$ (Table 3) and turbidity measured from
HACH and OBS (side-scattering measurements) ranges between 0.9 and 771 FNU and between 0.2 and 632 FNU
respectively. A very good relationship is observed between SPM and turbidity which almost follows the 1:1 line
for both instruments (Figure 3). A linear model between both parameters gives very good coefficients of determi-
nation ($R^2 = 0.98$ for HACH and $R^2 = 0.95$ for OBS) and slopes (0.92 for HACH and 0.86 for OBS). However, we
can notice that for very high turbidity (> 500 FNU), turbidity values measured by HACH tends be slightly higher
than SPM values (Figure 3A) but not OBS turbidity values. As expected from previous results, when comparing
side scattering turbidity obtained from OBS and turbidity measured by HACH, a good relationship is retrieved
(Figure 4A) with a $R^2$ of 0.96 and a slope of 0.84. Despite larger variability for very high turbidity, these results
confirm that OBS is a good tool for continuous measurements of turbidity in turbid environments.
The ratio of the side scattering versus the back scattering derived from OBS measurements has a particular interest
as it can provide information on the size and properties of the particles, e.g higher ratio could be explained by
larger particles (Nechad et al., 2016). In the HYPERMAQ dataset this ratio (Figure 4B) displayed a very high
variability in low turbidity environments and an increasing slope for high turbidity environments (i.e. Pauillac) as
also observed by Nechad et al. (2016). The very high variability when turbidity is low is explained by the strong
impact of uncertainty on low back scattering values in the ratio calculation. In Figure 4B, it can be observed that





the side scattering versus back scattering ratio varies significantly between and within sampled sites. For instance,
this ratio is higher in the Gironde Estuary at Le Verdon than in the Spuikom lagoon. It seems also to be higher in
the Río de la Plata and in the Gironde at Pauillac than in the Chascomús lake, though the Río de la Plata showed
high variability. Finally, the median ratio of the whole dataset is 1.77 which is close to the mean value of 1.72
found in Nechad et al. (2016) in turbid waters.

**4.2 Chl-a and other pigments concentrations**

[Chl-a] are extremely variable within HYPERMAQ test sites with values ranging between 0.91 mg m$^{-3}$ in the
Gironde Estuary at Le Verdon and 180.7 mg m$^{-3}$ in the Chascomùs lake, although most of the observations are
within the range of 3 mg m$^{-3}$ to 10 mg m$^{-3}$ (Table 4). In addition, very high variability is observed within Belgian
waters and Spuikom, with [Chl-a] values ranging by a factor 10. This variability is mainly due to the fact these
study areas have been sampled at two different seasons (i.e. spring and summer).
Phytoplankton pigments derived from HPLC analysis were available in the Gironde Estuary, in the Belgian coastal
waters and in the Spuikom. The relative contribution of some key pigments for phytoplankton groups identification
(Uitz et al., 2006; Mackey et al., 1996) are represented on Figure 5. In the Gironde Estuary, at Le Verdon, signif-
icant concentration of fucoxanthin, peridin and chlorophyll-b are observed suggesting that diatoms, dinoflagellates
and chlorophytes are co-existing at similar levels. However, at Pauillac where phytoplankton biomass is higher
(Table 4), the high concentration in fucoxanthin suggests that planktonic assemblage is dominated by diatoms. In
Belgian waters, high value of fucoxanthin is also observed. This pattern was expected as fucoxanthin characterized
the two phytoplankton groups which are dominant during spring and summer in the Southern North Sea: diatoms
and the prymnesiophyte *Phaeocystis globosa* (Lancelot et al., 2005). The last one is also characterized by the
presence of chlorophyll-c$_3$. In the Spuikom, fucoxanthin and chlorophyll-b show high concentrations indicating
an important proportion of diatoms and chlorophytes.

**4.3 Absorption and attenuation coefficients**

Very wide ranges of light absorption and attenuation coefficients were measured as representative of low to ex-
tremely turbid waters. As expected in CDOM- and sediment-rich waters, the spectral variations of the non-water
absorption coefficients were closely following an exponential function, with decreasing values from short visible
to near-infrared wavelengths (Figure 6A). The respective contributions of CDOM and suspended particles to light
absorption at 440 nm (Figure 7) were observed to vary from 20 % to 40% for CDOM and hence from 60 % to 80%
for suspended particles, which could be expected in productive waters strongly influenced by sediment inputs from
rivers and resuspension effects.
The spectral variations of the non-water attenuation coefficients (c$_{nw}$, Figure 6B) showed a smooth decrease with
increasing wavelengths, closely following the power-law function with varying slopes. These variations of the
spectral slope are expected to be representative of different particle size distributions due to the combined influ-
ences of wind-driven and tidal currents, and to the mixing between mineral-rich (sediments) and organic-rich
(phytoplankton) particles.



### 4.4 Water reflectance

The large diversity of water-leaving reflectance spectra is displayed in Figure 8. Maximum reflectance in each spectrum varies from less than 0.02 on some spectra of the Belgian coastal waters to more than 0.15 in the Gironde Estuary at Pauillac. Shapes of the spectra are also very variable. The mark of strong chlorophyll-a absorption around 670 nm is well observed in the Chascomús and Spuikom lakes as well as in some spectra of the Belgian coastal waters. The two extremely turbid sampling stations, the Rio de la Plata and the Gironde at Pauillac, show some similarities in their spectral shapes although a large variability is observed at Pauillac due to a larger impact of tides.

The relationship of water reflectance at 645 and 860 nm and turbidity (Figure 9) shows expected patterns with a saturation of the reflectance at 645 nm when turbidity is higher than 200 FNU. Indeed, for these extreme turbidity values the band at 860 nm shows a more linear relationship. These results also show the limits of the standard algorithm of Nechad et al. (2009) for high turbidity or in case of a different environment like the Chascomús lake which is characterized by high [Chl-a] and turbidity.

### 5. Conclusion

Coastal and inland waters strongly interact with human activities. Some of these activities, like fisheries or tourism, rely on a good ecological status whereas the same activities but also others like farming, industry or urbanization tend to affect water quality. Hence, monitoring these waters is extremely important and for that optical remote sensing is a valuable tool as it allows a large spatial and temporal coverage. However, it is still challenging to retrieve biogeochemical parameters in complex case 2 waters (Odermatt et al., 2012) because the transfer of light in water is affected by temporally and spatially variable inputs of CDOM and terrestrial sediments. To help the scientific community to build comprehensive database for the development of algorithms, the HYPERMAQ dataset provides data for seven different studies areas with SPM and [Chl-a] ranging from moderate to extremely turbid and productive, and located over three continents (i.e. Europe, South America and Asia). The HYPERMAQ dataset includes big river estuaries characterized by high turbidity, inland lagoons with productivity ranging from moderate to extreme and finally Belgian coastal waters in the North Sea characterized by the high spatio-temporal variability of optical properties (Vantrepotte et al., 2012). The parameters shared in the HYPERMAQ dataset include descriptors of biogeochemical conditions (i.e. [Chl-a], SPM, turbidity) as well as AOPs (i.e. water reflectance) and IOPs ($a_{nw}$ and $c_{nw}$). Although this dataset does not aim to cover the whole variability of case 2 waters, it provides valuable information to describe turbid and even extremely turbid waters and has the potential to help the development of remote sensing algorithms. It can also contribute to the production of a larger optical database, based on in situ measurements for a comprehensive description of case 2 waters.



**Data availability**

Data is available from Lavigne et al. (2022), hosted at PANGAEA (http://www.pangaea.de) under the doi: https://doi.pangaea.de/10.1594/PANGAEA.944313

**Author contributions**

HL, AD, DD, FC, AC, XS, JIG, RP, MB, QV and KR participated to one or more field campaigns. Data processing has been made by HL, AD and JIG (turbidity), DD (absorption), KR, MB, QV, RP, AD (water reflectance), AC, AD, DD, FC, KS (chlorophyll-a, pigments and SPM). HL, DD, AD have compiled data and created the final dataset. All authors participated to manuscript redaction and revision.

**Competing interests.**

The authors declare that they have no conflict of interest.

**Acknowledgements**

This work has been founded and promoted by the Research programme for earth observation 580 STEREO III HYPERMAQ project (contract nr SR/00/335). Flemish LifeWatch BE programme, funding by FWO, is thanks for its contribution to the water sampling in the Belgian Coastal Zone. We thank VLIZ for providing the Zeekat and shiptime on the RV Simon Stevin and her crew for their support during sampling. Inland water sampling in Belgium was also founded by the Belspo PONDER (SR/00/325) project. The SAPIGH analytical plateform of the "Institut de la Mer de Villefranche" (CNRS-France) is thanks for having performed the analysis of pigments in the Gironde. NASA, USGS, ESA and EUMETSAT are thanks to offer a free access to Landsat 8 and Sentinel 2 images.

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



**Table 1: Date, location and platform of the campaigns.**

| Campaign | Date | Platform | Latitude (deg) | Longitude (deg) |
|---|---|---|---|---|
| Spuikom | 19 April 2018, 23,24 and 27 July 2018 | Inflatable boat | 51.23 | 2.95 |
| Belgian Coastal waters | 23-25 April 2018, 25-26July2018 | RV Simon Stevin | 51.18-51.59 | 2.50-3.15 |
| Gironde (Pauillac) | 17 and 19 Sept. 2018 | Harbor | 45.1975 | -0.7422 |
| Gironde (Le Verdon) | 18 and 20 Sept. 2018 | Pier | 45.5438 | -1.042 |
| Chascomùs | 9-10 April 2018 | Pier | -35.5828 | -58.0202 |
| Rio de la Plata (Buenos Aires) | 4-5 April 2018 | Pier | -34.5609 | -58.3988 |
| Xangtze (Chongxi) | 31 May, 1 and 3 June 2018 | Pier | 31.759 | 121.193 |
| Xangtze (Baozhen) | 4-8 June 2018 | Pier | 31.520 | 121.609 |


**Table 2: Number of observations for each sampling site. *also include pigments concentrations from HPLC.**

| Campaign / Site | TriOS | TUR (HACH) | TUR (OBS) | $a_{nw}$ - $c_{nw}$ AC-9 | SPM | [Chl-a] |
|---|---|---|---|---|---|---|
| Spuikom | 27 | 27 | 23 | 11 | 17 | 17* |
| Belgian coastal waters | 18 | 19 | 17 | 10 | 19 | 18* |
| Gironde - Pauillac | 25 | 26 | 26 | 23 | 13 | 13* |
| Gironde – Le Verdon | 21 | 25 | 25 | 24 | 12 | 12* |
| Chascomùs | 5 | 5 | 5 | 5 | 5 | 3 |
| Rio de la Plata- BA | 16 | 22 | 22 | 18 | 10 | 10 |
| Xangtze - Chongxi | - | 19 | - | - | 17 | - |
| Xangtze - Baozhen | - | 37 | - | 29 | 37 | - |



**Table 3: Distribution of SPM (g m$^{-3}$) in each sampling site.**

| Campaign / Site | SPM (g m$^{-3}$) | | | |
|---|---|---|---|---|
| | min | median | mean | max |
| Spuikom | 2.06 | 3.16 | 3.93 | 8.40 |
| Belgian coastal waters | 1.02 | 4.49 | 9.63 | 62.04 |
| Gironde - Pauillac | 22.5 | 181 | 177 | 474 |
| Gironde – Le Verdon | 5.85 | 7.80 | 10.2 | 23.5 |
| Chascomús | 81.0 | 175 | 141 | 189 |
| Rio de la Plata | 49.3 | 71.7 | 74.0 | 93.8 |
| Xangtze - Chongxi | 27.2 | 42.2 | 44.8 | 66.4 |
| Xangtze - Baozhen | 23.6 | 52.8 | 53.6 | 138.4 |




**Table 4: Distribution of Chl-a concentration (mg m⁻³) in each sampling site.**

| Campaign / Site | Chl-a (mg m$^{-3}$) | | | |
|---|---|---|---|---|
| | min | median | mean | max |
| Spuikom | 2.40 | 9.16 | 10.64 | 22.70 |
| Belgian coastal waters | 1.99 | 6.33 | 7.49 | 17.36 |
| Gironde - Pauillac | 2.49 | 3.82 | 3.88 | 6.85 |
| Gironde – Le Verdon | 0.91 | 1.63 | 1.67 | 2.79 |
| Chascomús | 141.5 | 141.5 | 154.6 | 180.7 |
| Rio de la Plata-BA | 2.17 | 3.27 | 3.72 | 8.71 |
| Xangtze - Chongxi | - | - | - | - |
| Xangtze - Baozhen | - | - | - | - |








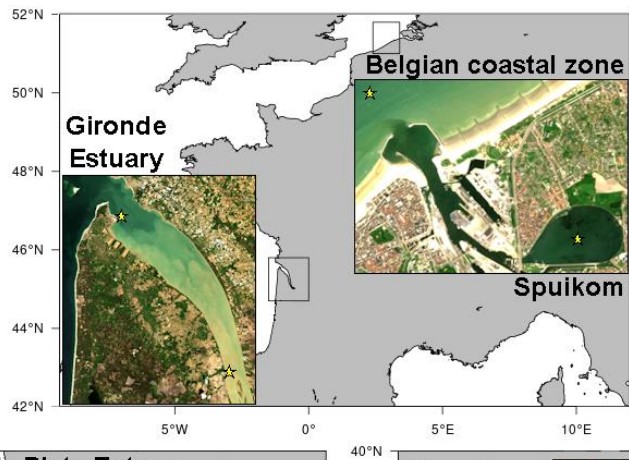

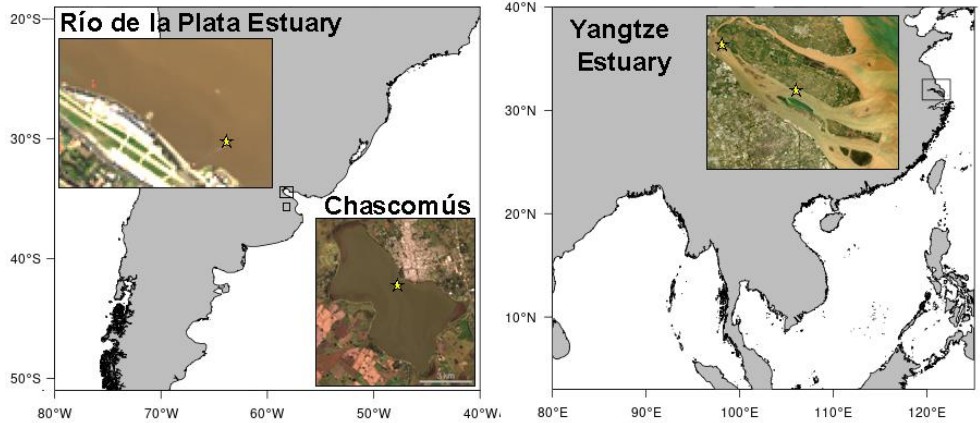


**Figure 1: Locations of the study areas. Satellite images are coming from Landsat 8 OLI (Yangtze: image taken on 2021-04-29, Chascomus: image taken on 10-05-2017, Rio de la Plata: image taken on 2014-08-13) and Sentinel 2B MSI (Belgian Coastal Zone: image taken on 2021-05-30, Gironde: image taken on 2021-05-03)**




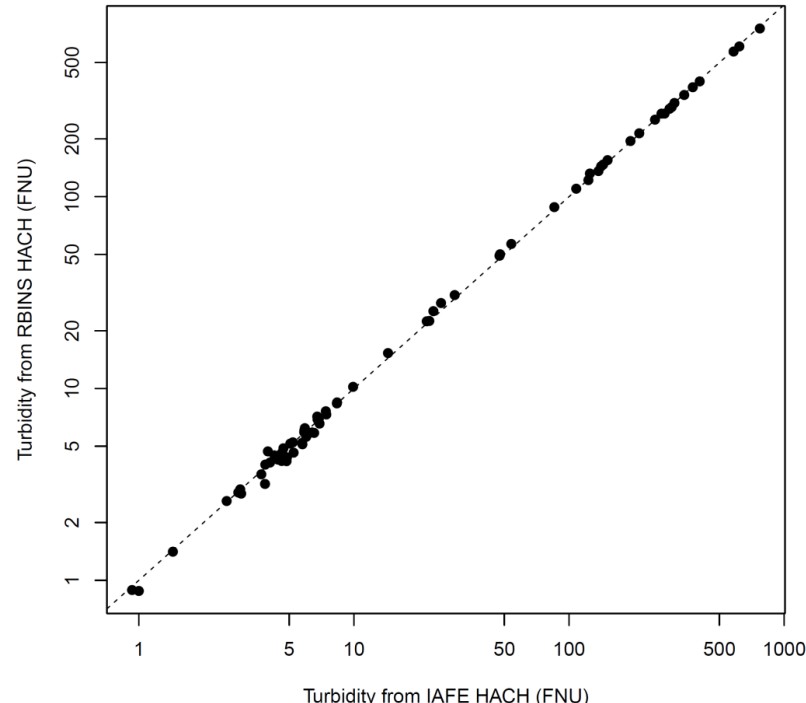


**Figure 2: Comparison of simultaneous measurements of turbidity made from two different HACH instruments ($r^2$=0.99).**





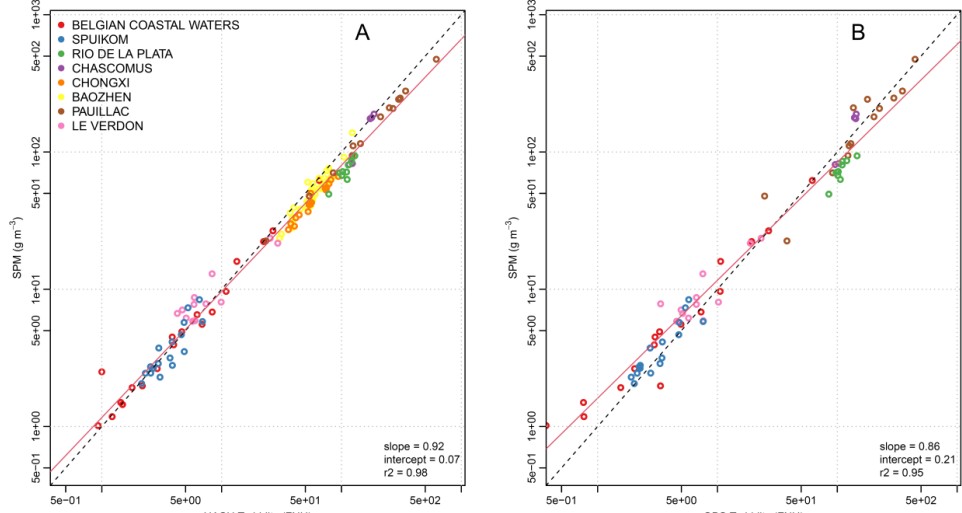


**Figure 3: SPM as a function of turbidity. Turbidity from OBS is given by the side-scattering measurement. The dotted line is the 1:1 line and the red line represents the linear regression between SPM and turbidity.**


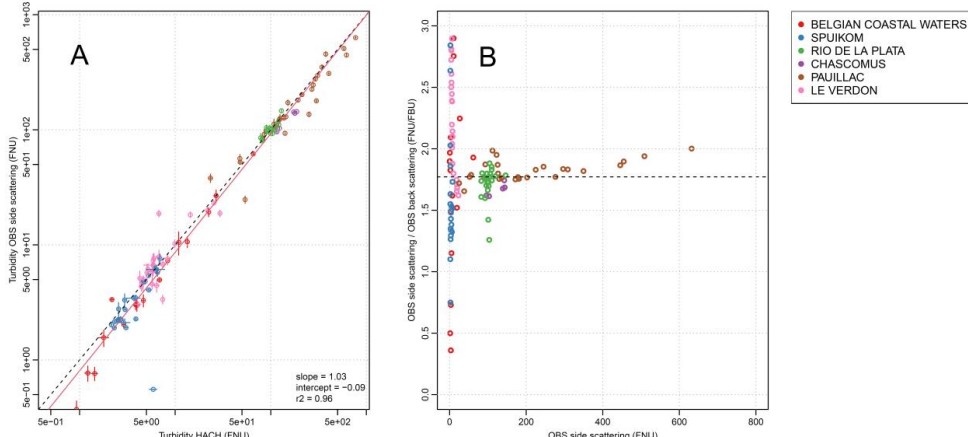

**Figure 4: Turbidity measured by HACH instrument as a function of side-scattering turbidity measured by the OBS instrument (panel A). The red line shows the least squares regression between these variables. Panel B: ratio of the OBS side-scattering to backscattering ratio as a function of the OBS side-scattering. The horizontal dotted line represents the median value of the scattering ratio.**



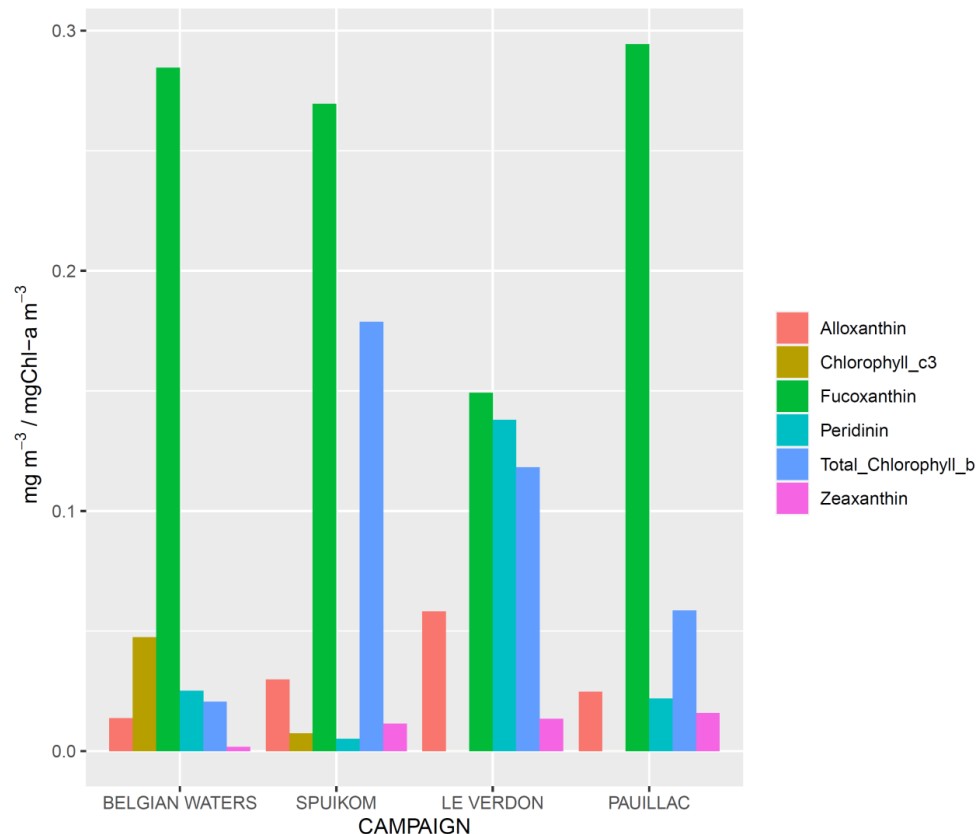

**Figure 5: For each campaign, average concentration of alloxanthin, fucoxanthin, peridinin, chlorophyll c₃, zeaxanthin and total chlorophyll b normalized by the concentration in chlorophyll-a**

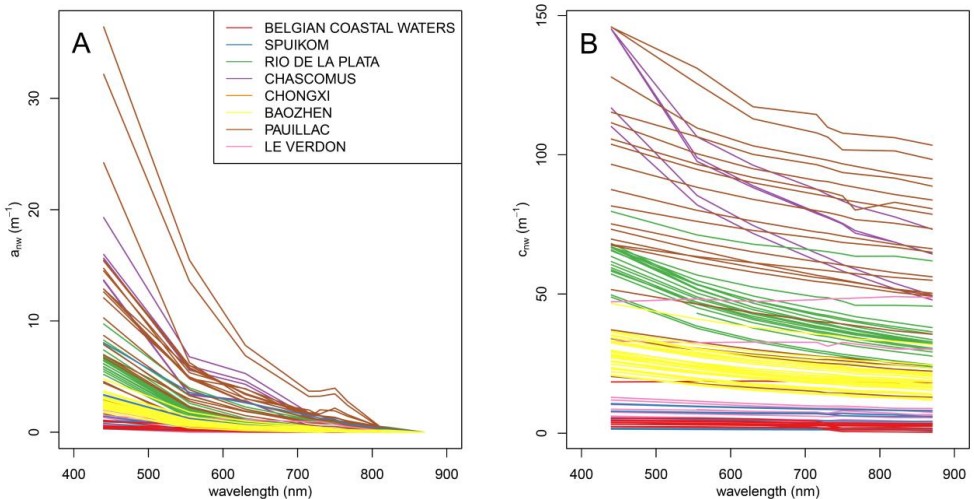


**Figure 6: Non-water absorption (panel A) and attenuation coefficients (panel B) measured with the AC-9 instrument.**


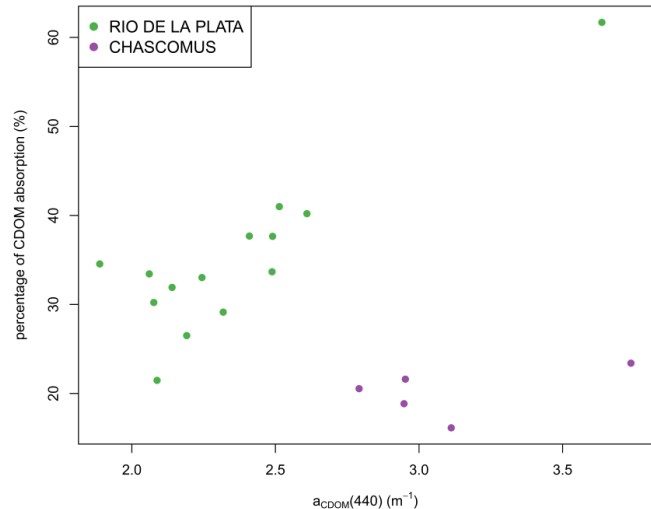

**Figure 7: Percentage of CDOM absorption as a function of $a_{CDOM}(440)$. $a_{CDOM}$ was only measured during two cam-**
**paigns.**

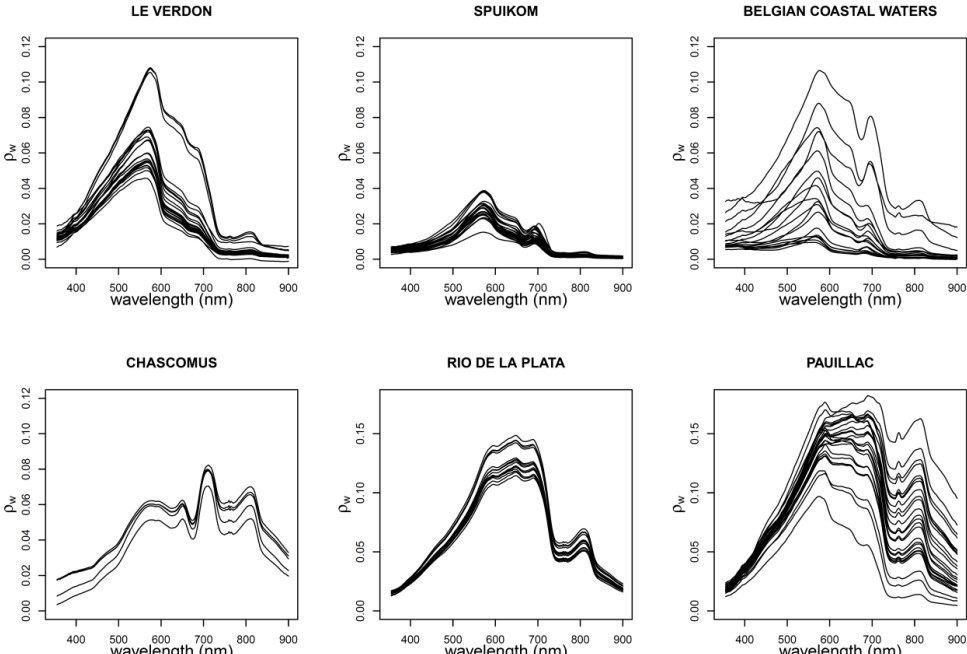


**Figure 8: Water reflectance spectra from each sample site.**



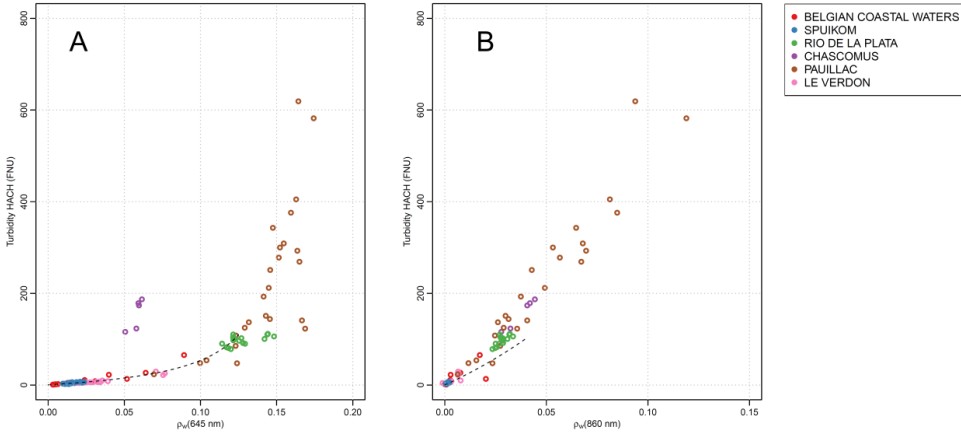


**Figure 9: Turbidity as a function of water reflectance at 645 nm (panel A) and 850 nm (panel B). Black dotted line
represents the model of Nechad et al. (2009) between 0 and 100 FNU.**