# Peer review of "The HYPERMAQ dataset: bio-optical properties of moder ately to extremely turbid waters"

_Earth System Science Data, 2022_

## Referee Comment (RC2)

[referee-annotated manuscript omitted]

---

## Author Response (AR1)

**Main changes in the manuscript:**

- A paragraph better describing the relevance of sampling sites has been added in Section 2.

- Figures 3, 4 and 9 have been updated with increased label size.
* * *
**Responses to Reviewer #1**

**We thank you for all the time you spent reviewing our manuscript and your constructive comments. In the following your comments are in black and our replies in blue.**

**General comments**

The manuscript proposed by H. Lavigne et al. aims to describe the HYPERMAQ dataset dedicated to provide optical and biogeochemical parameters necessary to develop algorithm for ocean colour observation to retrieve main biogeochemical parameter as the chlorophyll-a concentration. Dedicated field experiments have been driven in six contrasted regions.

The manuscript is concise, well written and details each HYPERMAQ dataset components.

As a general comment, I mainly suggest to add a discussion on the choice of the six sampled regions. Indeed, except a short justification in the conclusion, the reader can't easily appreciate the interest of sampling such different regions.

Reply: Indeed, the rational for this particular selection of sampling sites might not be very clear, the paragraph 2 has been extended to better explain the interest of these sampling sites.

Furthermore, this paper described a bundled dataset. Each individual dataset is also associated to Pangaea doi. It would be interesting to list (and associate DOIs) in the bundled paper.

Reply: Thank you for this suggestion, sub-datasets are now listed in the section "data availability"

Considering the limited needed improvements included in general and specific comments, I recommend this paper for potential publication after minor revisions.

**Specific comments**

Title

The title is built on a project/dataset name, maybe linked to an undefined acronym (found on the web ? Hyperspectral and multi-mission high resolution optical remote sensing of aquatic environments). I think that the title would benefit from a more explicit title giving an idea of the addressed dataset and/or parameter.

Reply: Thank you for your suggestion title has been changed to "The HYPERMAQ dataset: bio-optical properties of moderately to extremely turbid waters"

Tables 1 to 3

Xangtze name is used instead of Yangtze.

Reply: Thank you for noticing this typo it has been corrected.

3. Data collection

p. 5-6 / l. 164 – 184

Sections are switching from 3.2 to 3.4. Section 3.3 is missing.

Reply: Thank you for noticing this problem it has been corrected.

5. Conclusion

p. 9 / l. 305 - Seven different studies areas are mentioned in the conclusion whereas six regions are described in the paper.

Reply: Thank you raising this inconsistency. It might have been a confusion with the two sampling cruises in the Belgian coastal waters, but in fact, there are only 6 sampling sites. This has been corrected.

Figure 8

Y-axis should be explicitly defined including considered units.

Reply: water reflectance is explicitly defined in the text as "$\rho_w$" and is reminded in the figure caption. Water reflectance is unitless. This is now reminded in the figure caption.

**Responses to Reviewer #2**

**We thank you for all the time you spent reviewing our manuscript and your constructive comments. In the following your comments are in black and our replies in blue.**

**Review of "The HYPERMAQ dataset" by Lavigne et al**

by Giorgio Dall'Olmo (gdallolmo@ogs.it)

**General comments**

This manuscript describes a dataset of optical properties (both apparent and inherent) and co-located measurements of turbidity, phytoplankton pigment and suspended matter concentrations. The dataset focuses on turbid productive waters.

Overall the manuscript nicely supports the dataset by describing its main features. The dataset is unique as it has been collected in strongly turbid and productive water bodies in different continents. The dataset appears to be of high quality and so is the manuscript describing it.

Reply: Thank you for these positive comments.

**Specific comments**

I have very few and minor specific comments in the attached pdf.

Here are main comments of the attached pdf and our replies:

Have you thought about merging / submitting the HYPERMAQ dataset to the LIMNADES database? This would help minimising the fragmentation of datasets that forces users to retrieve data from different sources.

Reply: We agree that for certain types of application, users need data from one or two parameters and prefer databases that gather different sources. This is why we propose to submit our data to the copernicus ocean colour in situ database (OCDB) as it should become one of the major databases for bio-optics in the coming years. This integration should be performed within the next months

Chascomus lake: please, could you make sure you are reporting the correct number of significant figures?
Reply: All reported values are exactly as they have been published in the cited references (Diovisalvi et al. 2014, Pérez et al. 2011).
Diovisalvi, N., Salcedo Echeverry, G.E., Lagomarsino, L.&Zagarese, M.E.,Seasonal patterns and responses to an extreme climate event of rotifers community in a shallow eutrophic Pampean lake. Hydrobiologia 1 (1), 13, 2014.
Pérez, G.L., Llames, M.E., Lagomarsino, L., Zagarese, H., Seasonal variability of optical properties in a highly turbid lake (Laguna Chascomús, Argentina). Photochemistry and Photobiology, 87: 659–670, 2011.

Line 222 [Chl-a]
"[Chl-a] and phytoplankton pigments" have been replaced by 'Phytoplankton pigments including [Chl-a]"

Line 224: not clear how exactly the pigment concentrations were obtained: HPLC again?
Reply: As stated lines 222 to 224: "phytoplankton pigments were determined using High Performance Liquid Chromatography (HPLC) [...] in campaigns in the Belgian coastal waters, the Spuikom and the Gironde." Hence, in Belgian waters, measurements provided by the LifeWatchBE come from HPLC analysis. As it was mentioned just above, we do not think this is necessary to mention it again. To be more consistent between sentences lines 222-224 and line 224, "Belgian waters" in line 224 has been changed to "Belgian coastal waters".

Figure 3: labels are very small: consider increasing their sizes
Reply: Thank you for this suggestion, size of labels has been increased in Figures 3, 4 and 9.

Figure 61: could it be better to use a log10 scale for the y axes?
Reply: We agree that log10 scale allows to better emphasis small values which are more numerous in our dataset of a_nw. However, for consistency and comparison with Figure 6B (c_nw) where log-scale is not necessary we prefer to keep a linear scale in Figure 6A.

Figure 8: use a same y range?
Reply: given the high diversity of water reflectance spectra shown on Figure 8, we have chosen to keep the same range (0-0.12) for all the sampling sites except the most turbid ones (Rio de la Plata and Gironde, Pauillac) where the y-range had to be slightly extended.

Lines 292: add reference
Reply: the reference Luo et al., (2018) has been added.
Luo, Y., Doxaran, D., Ruddick, K., Shen, F., Gentili, B., Yan, L., & Huang, H. (2018). Saturation of water reflectance in extremely turbid media based on field measurements, satellite data and bio-optical modelling. *Optics express*, *26*(8), 10435-10451.